# Effects of Hydrologic Regime Changes on a Taxonomic and Functional Trait Structure of Earthworm Communities in Mountain Wetlands

**DOI:** 10.3390/biology12030482

**Published:** 2023-03-21

**Authors:** Václav Pižl, Maria Sterzyńska, Karel Tajovský, Josef Starý, Paweł Nicia, Paweł Zadrożny, Romualda Bejger

**Affiliations:** 1Institute of Soil Biology and Biogeochemistry, Biology Centre CAS, Na Sádkách 7, 370 05 České Budějovice, Czech Republic; pizl@upb.cas.cz (V.P.); tajov@upb.cas.cz (K.T.); jstary@upb.cas.cz (J.S.); 2Museum and Institute of Zoology PAS, Wilcza 64, 00-679 Warsaw, Poland; 3Department of Soil Science and Agrophysics, University of Agriculture in Krakow, Al. Mickiewicza 21, 31-120 Kraków, Poland; rrnicia@cyf-kr.edu.pl (P.N.); pawel.zadrozny@urk.edu.pl (P.Z.); 4Department of Bioengineering, West Pomeranian University of Technology in Szczecin, Papieża Pawła, VI no 3, 71-459 Szczecin, Poland; romualda.bejger@zut.edu.pl

**Keywords:** environmental change, disturbance, biodiversity, functional traits, resilience, resistance, earthworms

## Abstract

**Simple Summary:**

Disturbance mechanisms, both natural and human-induced, can lead to changes at all levels of biological organization, including soil biota. Understanding how soil biota has adapted to a dynamic system driven by natural disturbances responding to environmental change requires a taxonomic and functional approach. Although wetlands are naturally hydrologic disturbance-driven ecosystems, relatively little attention has been paid to soil biota responses to environmental changes in these disturbance-dependent ecosystems. It has been hypothesized that the drainage-related changes in mountain wetlands soils have an effect on the community composition and functional structure of soil biota. For the first time, a field study has demonstrated that hydrologic disturbances affect the functional and taxonomic composition of soil biota represented by earthworms. Abiotic-based environmental filtering was responsible for the earthworm species sorting and trait variation. The highest earthworm variation found in wetlands semi-natural hydrologic conditions emphasizes the impact of transient dynamics on earthworm assembly.

**Abstract:**

Disturbances, both natural and anthropogenic, influence the patterning of species and species traits. The shift in species composition and distribution pattern of functional traits can demonstrate if the community is resistant, sensitive or resilient to the disturbance. Based on species- and trait-based approaches, we examined the response of the earthworm community to changing hydrologic conditions caused by the artificial drainage of mountain fens, in which cumulative effects of disturbance events over space and time are much less dynamic than in riverine wetlands. We hypothesized that the drainage-related changes of mountain fen peat soils have an effect on the earthworm community composition and its functional structure. We assume that the shift in species composition and value of community-weighted functional traits reflect changes in the resilience or resistance of the earthworm community to environmental change. Our results demonstrate that the total density of earthworms was almost three times lower under drained conditions compared to natural ones. Artificial drainage of fens had a neutral effect on the species-based diversity indices. However, there were species-specific traits that responded to hydrologic changes and which led to the species’ replacements and to the co-occurrence of eurytopic, surface-browsing and more drought- and low-pH-resistant earthworm species in the drained fens. Based on these results, we conclude that abiotic-based environmental filtering was the main process responsible for sorting earthworms according to species and traits in the disturbed hydrologic conditions. The greater earthworm functional trait variations in semi-natural hydrologic conditions emphasizes the impact of transient dynamics in an altered disturbance regime on the earthworm assembly. Results also showed that in the changing hydrologic conditions of mountain fens, the functional trait approach revealed only slightly more predictive power than the taxonomic one, but it proved better with processes responsible for earthworm species filtering.

## 1. Introduction

Disturbances affect all levels of biological organization [1,2] and often are the main factor that influences the patterning of species and species traits within a given community [3]. There is a wide variety of literature available on the effects of natural and human-driven disturbances on the soil invertebrate community structure [4]; however, relatively scarce attention has been given to the response of soil invertebrates to environmental changes in naturally disturbance-driven dynamic ecosystems such as wetlands [5,6,7,8,9].

The hydrologic regime in wetland ecosystems is the primary driver structuring the physical habitat, providing habitat connectivity and framing biotic interactions [10,11]. The cumulative effects of hydrologic regime dynamics over space and time and the abiotic stress mainly imposed by periodic flooding of the soil, submergence occurring in deeper soil horizons and normal drought events lead to different adaptations and predispositions of soil invertebrates at the individual, population and community level [12,13,14]. Intensity and frequency of the disturbances are mainly responsible for the extent to which they modify the soil biota community composition and associated pool of species traits, which in turn have an effect on community resilience and resistance [15]. Therefore, to improve our understanding of how the soil biota community, which is adapted to a naturally disturbance-driven system, changes under the shifting of the disturbance regime requires a taxonomic and functional approach [16,17]. Combining these two approaches, which define species in terms of diversity patterns and their ecological roles and interactions with the environment, allows us to scale up from the species to the community level. Functional attributes have a highly predictive value and the potential to provide evidence on the mechanisms of how the structure of local communities respond to natural and human-driven disturbances [18,19,20]. They can determine the biological community response to environmental change better than the taxonomic approach [17], and they are helpful in identifying environmental filters and strategies that permit species to pass through all environmental filters [21]. A trait-based approach allows us to understand links between filtering factors and functional traits in the system [22], and it can also offer fundamental insights into ecosystem resilience and resistance to a changing disturbance regime [23,24,25]. Within a functional trait framework, comprising response-and-effect traits [17], community-weighted response traits are those that are relevant to scale the community response to environmental change, while effect traits reflect the change of ecosystem processes [26].

The trait-based approach is still much less advanced in the ecology of soil invertebrates than in plant ecology [27]. Nevertheless, available studies have demonstrated that the range of functional variations in soil invertebrate communities may be of high value in land use change [18], climate manipulation [28], recovery patterns after fire [29], soil contamination [30] or the determination of mineral and organic N dynamics in a soil plant system [31]. Relatively little attention has been paid to the assessment of how the alteration of the natural disturbance regime changes the trait distribution and community assembly of soil invertebrates, and to the positioning of their response in the context of the ecological resilience concept [7,9]. The grouping of soil invertebrates by functional traits as a proxy for the evaluation of ecological resilience has been applied entirely to earthworm communities under different intensities of inundation as a stressor in flood-prone areas within a river–floodplain system [7]. However, wetland ecosystems are highly variable, and their natural conditions are controlled by different hydrodynamic behaviors related to various degrees of vertical water-table fluctuation and the rate of lateral groundwater flow. Wetland sites can be characterized by a very dynamic soil water regime, such as those within the river floodplain; in addition, they may include sites without obvious surface flooding and with a stable water regime, such as bogs and fens. In wetland ecosystems, besides water stress and differences in hydrologic regime dynamics, natural and human-induced disturbances act in conjunction with various combinations of hydrological and geomorphological characteristics. These define various physical/hydrological habitat templates [32,33] which can influence the species persistence and community structure of the residing biota in a different way, implying different experiences of a community to a given range of disturbance regimes (sensu Keane [34]). In addition, local conditions under which the disturbance occurs are also important as they affect the taxonomic and functional composition of biotic communities [35], modification of the pool of species traits and their resilience or resistance to a disturbance [36].

Slope wetlands with constant groundwater seepage have unique hydro-ecological functions and vulnerabilities to environmental change, different from those encountered within riverine floodplains [37]. Among them, mountain fens have specific climatic, geological, geomorphological, hydrographical and hydrogeological conditions [38,39]. Nevertheless, as in other types of wetlands, the hydrologic regime is the main environmental driving force, and it represents a good proxy for the complex hydrologic–edaphic gradients associated with the decline of the groundwater level. In Europe, many mountain fens have been subjected to artificial drainage for centuries to make them accessible for the forest industry. This has caused changes in hydrologic properties [40] and in soil-forming processes [41,42,43]. The long-term drier conditions in peatlands induced by drainage, by analogy, might represent potential effects of climate change under air warming, not only on the hydrologic properties and processes [44] but also on the soil fauna response. There is little research on soil fauna in forested wetlands in mountainous regions, indicating a disregard for the fact that soil invertebrates are important in many soil processes and represent useful bioindicators of human disturbances [45]. Moreover, in spite of the overall effect produced by hydrologic conditions (HC), changes in mountain fens have a strong impact on soil properties and the response of individual soil macro-decomposer components, as demonstrated for millipedes and terrestrial isopods [9].

Earthworms are known as ecosystem engineers, demonstrating a substantial impact on soil functioning and ecosystem services such as soil formation, nutrient cycling, primary production and water regulation [46]. They can offer an important tool to evaluate different environmental transformations and impacts [45,47] as well as the effectiveness of restoration measures.

To our knowledge, little is known about earthworm taxonomic and functional community responses to environmental changes in mountain fens, in which the cumulative effect of disturbance events over space and time is less dynamic than in riverine wetlands. We hypothesized that drainage-related changes in mountain fen peat soils have an effect on earthworm community composition and functional structure. We assume that the shift in species composition and the value of community-weighted functional traits of species reflect changes in the resilience or resistance of the earthworm community. To test this hypothesis, we performed our study in naturally dynamic mountain fens without obvious surface flooding and with stable water regimes but varying in the degree of hydrologic regime changes triggered by artificial drainage: natural (without artificial drainage), semi-natural (drained, but with no active drainage system) and degraded (drained).

The objectives of the present study were (1) to quantify the response of earthworm community composition and community-weighted functional traits of species to hydrologic changes), (2) to evaluate which functional traits contribute to the prediction of hydrologic changes, and (3) to assess the relationship between patterns of earthworm communities, community-weighted means of earthworm traits and environmental variables.

## 2. Materials and Methods

### 2.1. Study Area

The study was carried out in fens of the lower submontane and montane zones of the Babia Góra Massif in the Outer Flysch Carpathians, the Central European Highlands biogeographic province, in a biome with mixed mountain and highland ecosystems of complex zonation. Since 1954, this area has been protected as the Babiogórski National Park (BNP), and in 1977, it was listed by UNESCO as a World Biosphere Reserve. The study was carried out in the mountain fens formed on the northern slopes with hardly permeable mineral layers and with a soligenous type of hydrological feeding. Such hydrogenic sites often represent the *Caltho-Alnetum* azonal forests (gray-alder–bog forests), the priority biotope listed in the annex to the 1st EU Habitats Directive. A characteristic feature of non-degraded mountain fens is the large accumulation of organic matter and water saturation in the soil profile. One cubic meter of fen organic soil retains between 300 and 900 dcm^3^ of water. Prior to legal protection, the majority of forest fens in the area of the Babiogórski National Park were drained in the 1970s in order to increase timber production. Drainage works triggered significant changes in the hydro-ecological conditions of the mountain fens and led to degradation involving muck-forming processes. For a detailed description of the study area, see Sterzyńska et al. [9].

### 2.2. Sites

Nine study sites of different degrees of hydrologic changes (Figure 1) were established in the *Caltho-Alnetum* stands (49°35′–49°37′ N, 19°31′–19°35′ E) on the mountain slopes. Three sites were selected with preserved natural hydrology and natural-like HC, without artificial drainage, fed by groundwater flowing out of aquifer outlets; three semi-natural sites were treated as representatives of transient dynamics in hydrologic changes, with past drainage and drainage ditches present but no longer used; and three degraded sites represented active drainage systems, treated as sites with an altered hydrologic regime. Further characteristics of the sites are given in Table 1.

### 2.3. Sampling and Processing

Each study site was sampled on six occasions: 2 June and 28 September 2010, 7 June and 5 October 2011, and 29 May 2012 and 9 September 2012. Three random soil blocks from the top layer (each 1/16 m^2^ in area and 10 cm deep) were hand dug at each site and on each sampling. In addition, below the upper soil sample, the immediately deeper soil layer was dug down to a depth of approximately 20–30 cm. Earthworms living in the deeper layer were hand-sorted on site. Earthworms from the top layer were transported to the laboratory and heat-extracted using the modified Kempson extraction apparatus for a minimum of 10 days. All identified specimens were deposited in the collection of the Institute of Soil Biology, Biology Centre CAS (České Budějovice, Czech Republic).

Physicochemical soil properties, such as hydrolytic acidity (H+), Mg^2+^ concentration, pH, electrical conductivity (EC), total exchangeable base cations (TEB), base saturation (BS) and cation exchange capacity (CEC), were measured and determined in separate soil samples in 2009–2012. The liming Mg^2+^ cation concentration was determined by atomic emission spectrometry. Hydrolytic acidity (H^+^) was analyzed using soil extraction by Na-acetate. Soil pH was measured potentiometrically, and soil EC was estimated using a conductometer. The TEB was determined as the sum of base cation concentrations (Ca^2+^, Mg^2+^, K^+^ and Na^+^). The calculations of CEC and BS percentages were performed by indirect estimation as follows: CEC = TEB + H+ and BS = TEB/CEC × 100%. Detailed descriptions of the procedures for analyzing physico-chemical soil properties were presented in an earlier paper [9]. Groundwater levels (G.W.L.) were monitored using PVC piezometers installed permanently in the middle of each study site, and the distance between the piezometer water level and the reference point was checked at the time of soil sampling.

### 2.4. Functional Traits

We used five biological, two performance and five ecological preference traits, which can respond to hydrologic changes (Table 2 and Appendix A). All five biological traits (length min, length max, prostomium, dispersal potential, quiescence) were based on morphology, i.e., maximum and minimum body length and position of the prostomium (chemical detector), and life history traits, such as dispersion and alternation between active and dormant stages related to seasonal, non-diapause dormancy (quiescence). We assumed that offspring output related to the number of cocoons per year as a component of fitness can also be correlated with soil moisture dynamics and groundwater levels. The selected ecological traits, such as hygrophily tolerance (adaptation to moist habitats), habitat width, acid-tolerance and C/N ratio preference, which result from the optimum distribution of a trait along an environmental gradient, may also respond to the effect of hydrologic changes in a mountain fen caused by drainage. Parameter values for these traits were obtained from the literature (e.g., [49,50]), the earthworm species database [51], DriloBASE [52] and Edaphobase [53].

To determine whether alterations in the hydrology of mountain fens induced changes in earthworm trait composition, the community-weighted mean trait scores (Tm) were calculated according to Garnier et al. [54] and Makkonen et al. [28], who applied it to soil invertebrates as follows:$$\mathrm{Tm}=\sum_{i=1}^{n}\mathrm{pixi}$$
where x = trait attribute of the ith species and pi = a relative density of the ith species.

### 2.5. Data Analysis

A univariate analysis, namely, the Kruskal–Wallis one-way ANOVA, was used to compare the soil properties of the upper layer and basic characteristics of the earthworm community structure, such as density (A individuals m^−2^), species richness per site (S), Shannon’s diversity index (H’) and Pielou’s evenness index (J’) at different hydrologic conditions (HC). The significance of differences was determined with a multiple comparison test of mean ranks (Tukey‘s HSD), applied after the Kruskal–Wallis ANOVA. Nested two-way ANOVA, as a part of the general linear model (GLM), was used to assess the separate effects of HC and fen site on the earthworm species density and community metrics such as total density, species richness, diversity and evenness. We assume that the interaction among HC and fen site due to heterogeneity among sites is negligible. In the analysis, we used HC as a fixed factor and site as a random factor (sites have been nested within HC). Because of the non-normal distribution of most species data and structural attributes, all density and species richness data were log_10_(x + 1)-transformed before analysis. Data for diversity (H’) and evenness (J’) were not transformed. The significance of differences between means was calculated post hoc by Tukey’s HSD test. Bartlett’s, Cochran’s and Hartley’s tests for the presence of homogeneity of variance and the Kolmogorov–Smirnov test for normality with Lilliefors correction were used before analysis. In situations where data assumptions for nested ANOVA were not fulfilled, only the HC effect was calculated with the Kruskal–Wallis test statistics.

The linear method (RDA) was employed to test the species- and trait-based responses of earthworm communities to HC changes in mountain fens. We calculated the RDA constrained by the hydrologic condition HC (natural, semi-natural and degraded); groundwater level G.W.L.; local topography LT (slope, elevation) and time T (season—autumn and spring; year—2010, 2011 and 2012). The linear RDA method was used because, in our case, the value of the gradient length calculated for the first axis of DCA was shorter than 3 SD (see Lepš and Šmilauer [55]). Variation partitioning by a partial redundancy analysis (pRDA) was performed to reflect the relative importance of the study factors HC, LT and time as a group of predictors. The pure effect of a particular factor was assessed when the remaining factors were removed and used as covariates. This allowed the effect of a given factor to be interpreted independently from interactions with other factors [56]. The shared effect was quantified as the difference between the independent effect of an environmental factor, without removing the effect of another factor, and the pure effect of this factor. The automatic procedure for statistical model selection of environmental variables was used to assess the potential value of each variable separately (marginal effect) and the partial (conditional) effect for predicting the model to explain the highest amount of variability in earthworm species composition. The significance of the models was estimated by the unrestricted Monte Carlo permutation test. Before calculation, all data were standardized per m^2^ basis, and abundance data for each species were log_10_(x + 1)-transformed prior to ordination. Furthermore, the GLM modelling of each trait was performed using sample scores from RDA axis 1 and axis 2 as explanatory variables with the predictor variable used in the form of a second-order polynomial; the quadratic function and the log link function, respectively. The GLM function in CANOCO was applied to test the statistical significance between earthworm trait patterns and those predicted by hydrological changes. In the taxonomic approach, we performed the univariate and multivariate analyses using the data matrix with *species x fen site* with the row data (individuals summed from 3 soil samples) collected at each of the fen sites during the same sampling occasion and then standardized to individuals per square meter and averaged per sample. In the trait-based approach, we performed the analyses using the data matrix with *traits x fen site.* In the analyses, time (season—spring and autumn; year of study—2010, 2011 and 2012) was treated as repeated measures for adjusting *p*-values as a function of correlation in the data.

The level of significance in all analyses was set at α = 0.05. Calculations were made using the Statistica 10.0 and CANOCO 4.5 software packages (with terminology after Lepš and Šmilauer [55]).

## 3. Results

### 3.1. Soil Properties

Different soil types were found among mountain fens. Sapric Rheic Histosol (Eutric), an accumulation phase of the peat-forming process occurring in the surface horizons of the natural and semi-natural fen stands, and the Sapric Histosol (Eutric Drainic and Dystric Drainic) were the predominant soil types in the degraded fen stands with a mucking process occurring in the surface horizon. Significant differences in soil chemical properties such as EC, hydrolytic acidity, Mg^2+^ cation concentration, TEB and BS were found between the mountain fens exhibiting different levels of hydrologic changes (Table 3). The soils from semi-natural and degraded fens were less fertile and more acidified. Significant increases in EC, hydrolytic acidity, TEB and BS and decreases in the concentrations of such cations as Mg^2+^ in fen soils were found (Table 3).

### 3.2. Earthworm Diversity Pattern and Density

A total of 1649 earthworms representing 10 species were identified across all nine mountain fens under the different HC (Table 4 and Appendix A). Total earthworm density monotonically decreased toward degraded fen sites where their density was significantly reduced by 35%. The effect of fen sites nested within different HC was significant for total earthworm density. The species richness and diversity indices (H’ and J’) were not affected by hydrologic changes or by the site effect nested within the different HC.

### 3.3. Earthworm Taxonomic Composition Pattern

A significant effect of HC was found only for four earthworm species: *Eiseniella tetraedra*, *Lumbricus rubellus*, *Octodrilus argoviensis* and *Octolasion tyrtaeum* (Table 4). The density of *O. tyrtaeum* and *E. tetraedra* decreased significantly in degraded fen sites, while the density of *L. rubellus* increased. The density of *E. tetraedra* and *O. argoviensis* had the highest mean density in the semi-natural conditions. Two earthworm species, *L. rubellus* and *O. tyrtaeum*, showed a site effect nested within different HC.

The significant effect of different HC on earthworm community composition was also confirmed by pRDA analysis. The pure effect of HC, although significant according to the Monte Carlo permutation test for all canonical axes (λtrace = 0.47, F = 3.193, *p* = 0.002), was equal to the variation accounting for LT (λtrace = 0.147, F = 3.207, *p* = 0.002) and explained 14.7% of the variation (Figure 2).

Our results also showed that all canonical RDA axes, constrained by HC, LT and time, explained 32.6% of the earthworm community composition variance in mountain fens (λtrace = 0.326, F = 2.363, *p* = 0.002). However, partitioning the variation into time (using HC and LT as covariates) indicated a non-significant effect of time (season and year of study) on the earthworm community composition variation (λtrace = 0.061, F = 1.316, *p* = 0.176). The overlap between models (shared variation) was low and gave negative values, indicating that the effect of HC and LT together was stronger than the sum of the two separate effects [57,58]. Forward stepwise selection of the environmental variables considered in the RDA analyses confirmed this result, indicating that size of the fen and location above sea level from the LT group of predictors and groundwater level from the HC group of predictors were the main predictive variables that explained a significant amount of the variation in earthworm community taxonomic composition (Table 5).

The correlation of earthworm species with different HC, in the case when LT and time were used as covariates, showed a shift in the distribution pattern of two species, *O. argoviensis* and *L. rubellus*, compared with the results obtained by nested ANOVA. The occurrence of *O. argoviensis* has been negatively correlated with natural HC, while that of *L. rubellus* has not been so evidently correlated with degraded HC (Figure 3).

### 3.4. Earthworm Trait Composition Pattern

The distribution pattern of the community-weighted trait mean scores (Tm) showed significantly increased values of maximum body length and habitat width but decreased hygrophily tolerance in the degraded HC and low pH tolerance in semi-natural conditions. Five traits, attributed to vertical distribution, non-diapause dormancy (quiescence), habitat width and hydrophilic and C/N soil preference, showed a site effect nested within HC, indicating the possible effect of site-specific conditions (Table 6).

The total variation explained by the RDA model, accounting for trait-based results that included the predictors HC, LT and time, was 37.2% (λtrace = 0.372; F = 2.892; *p* = 0.002) and was only slightly higher than those accounting for species-based results. The variation uniquely attributable to HC (λtrace = 0.137; F = 3.206, *p* = 0.004) and LT (λtrace = 0.136; F = 3.170; *p* = 0.010) explained a similar percentage of variation among earthworm traits, but time also had a significant effect (λtrace = 0.120, F = 2.798, *p* = 0.016). The achieved negative amount of shared variation indicated that the joint effect of these predictors on community variation was stronger than the sum of the separate effects of both predictors [57,58] (Figure 4).

The main environmental variables predicting trait variation among earthworm communities were related to semi-natural fen HC conditions, elevation, size and time (2010 year) (conditional effect; Table 7).

In the case of the pRDA model constrained by HC (using LT and time as covariates), the first and second RDA axes were significant (λ1 = 0.233, F = 13.356, *p* = 0.006; λ2 = 0.147, F = 9.674, *p* = 0.004, respectively). This was essentially driven by the decrease in groundwater level and associated with changes from natural to semi-natural conditions along the first RDA axis and with degraded conditions for the second axis. The observed changes in HC and groundwater level along the RDA axes were correlated with changes in most earthworm trait values and demonstrated that the degraded conditions favored earthworm species with tanylobic prostomium, higher habitat width (eurytopic), low tolerance for hygrophily, tolerance for low pH and a high C/N soil preference. The natural sites favor species with a dormant stage (quiescence) and longer time to maturity (>52 weeks), (Figure 5).

The GLM models, run on the earthworms’ community-weighted mean trait scores (Tm) and the sample scores from the first canonical RDA axis, were constrained by the decreasing groundwater levels and changes from natural to semi-natural conditions. Using LT and time as covariates, these models showed that maximum length, dispersal potential and quiescence from biological traits, number of cocoons from performance traits, vertical distribution, habitat width and hygrophily tolerance from ecological traits were significantly correlated with axis 1. Traits such as the type of prostomium, habitat width and hydrophilicity tolerance showed a positive correlation with the second RDA axis constrained by degraded conditions, except for low pH tolerance trait, which presented a negative Tm score (Table 8).

## 4. Discussion

### 4.1. The Effects of Artificial Drainage (Hydrologic Changes and Local Topography) on Earthworm Community Structure and Composition–Taxonomic Approach

Forestry-related drainage of the mountain peat soils usually influences almost all components of matter balance [40,59], shifts the direction of the soil-forming processes [40,41] and induces changes in soil macro-decomposer communities [9]. This study documented that the communities of earthworms are also modified by the artificial drainage of mountain peat soils. The observed decrease in earthworm density seems to be bound with the successive disappearance of alder trees from mountain wetlands as an effect of drainage [60]. It seems that the falling out of alder trees from degraded fen stands and triggered changes in litter residues may alter both the quality and quantity of litter and soil organic matter, among others, through the release of nitrogen fixed by alder, and consequently, induce changes in the response of earthworm populations to N sources as it was assumed by Butt et al. [61] and Andriuzzi et al. [31]. Nevertheless, local earthworm movement could also respond to the trade-off between food availability and soil water conditions [62]. Aside from changes in the spatial and seasonal distribution of the organic matter pool on which earthworms actually live [63,64], soil moisture scarcity [65], as well as soil acidity and factors related to pH such as CEC [66,67] may also represent the possible limiting factors influencing the spatiotemporal distribution pattern and density of earthworms in drained conditions.

Contrary to expectations, the drainage of mountain fens did not significantly affect the taxonomically based diversity indices, which may indicate the random (neutral) processes in shaping earthworm responses to hydrologic regime changes. Despite this, four of ten species recorded (*Eiseniella tetraedra*, *Lumbricus rubellus*, *Octodrilus argoviensis* and *Octolasion tyrtaeum*) showed significant responses to hydrologic changes. However, their responses differed and were independent of the life form to which they had been a priori classified [65,68]. Significant decreases in the density of amphibious *E. tetraedra* (epigeic) and hygrophilous *O. tyrtaeum* (endogeic) in drained sites compared to permanently water-saturated soils of natural fens confirmed that they are susceptible to groundwater withdrawal [69]. On the other hand, epi-endogeic *L. rubellus* showed a significant increase in density in the drained sites and thus, its sensitivity to water-saturated conditions. The distribution pattern of *O. argoviensis* and the hump-shaped model detected with increasing density in semi-natural sites indicates that the avoidance behavior to highly saturated versus drained conditions could also be used to examine the range of environmental changes in mountain fens. Further, differences among the fen sites nested within different HC had also a significant effect on the density of *O. tyrtaeum* and *L. rubellus*. The observed differences in the density of *O. tyrtaeum* among fen sites could be related to the presence of barely morphologically distinguishable genetically different lineages of this taxon, which could differ in their ecological preferences [70] and thus increase the variability of *O. tyrtaeum* density within the investigated sites. Nevertheless, most individuals were large [71], with brownish bodies, and they always possessed glandular tumescences on the 22nd segment. This group has a much stronger preference for moist soils than the eurytopic group, individuals of which are whitish in color, at least in central Europe (Pižl, personal observation). On the other hand, *L. rubellus* is a successful colonizer of various natural and cultivated soils [64] and the viability of its local population is constrained by site-specific conditions, including the length of the dry period [72].

Variance partitioning analysis showed that a relatively large part of the earthworm community composition variation was related to the shared effect of the examined predictors, such as HC and LT. Our findings were consistent with the view that environmental factors, including the local topography, are important forces in shaping community composition and species diversity patterns [73]. The results of variance partitioning have been supported by forward selection analysis, which showed that fen patch size, elevation and groundwater level are important factors shaping the composition of earthworm community in mountain fens. This corresponds with the fact that wetlands in mountain areas often occur as isolated habitat patches scattered throughout the landscape matrix (mainly spruce and beech forests in our study), whose size, shape, location and orientation depend on the land relief [38,74], type of hydrological water feeding and surface runoff [60].

The effects of fen patch size and elevation on the earthworm community variation revealed in this study can be attributed to local habitat geometric parameters and may drive colonization-extinction dynamics of earthworms [75]. The earthworm community variation might also be linked to habitat structural complexity which could be independent of the effect of the patch size [76], or it may be greater within larger habitat patches as a result of increased spatial structure in the physical environment [77].

Within mountain wetlands, the broader dynamics of heterogeneity related to increased variability in micro-habitats formed by variations in micro-topography, the presence of more edges, and/or higher variability in soil properties [78] provide greater diversity of niches. This could affect the key parameters of metapopulations (i.e., extinction and colonization), landscape permeability and species dispersion capability, which facilitate movement among suitable habitat and resource patches. Many organisms experience their landscape as a mosaic of patches that vary in quality. However, the size and quality of habitat patches, the distances between them and the biotic and abiotic conditions in the intervening matrix habitat can affect their dispersion [79]. It has been identified that the earthworm population density is associated with habitat quality and land management practices (e.g., [66]), and the local movement of earthworms responds to a trade-off between food availability and soil water conditions [62]. On the other hand, earthworm species displayed a patchy distribution with different numbers of clusters and gaps, and the size of population clusters was different for epigeic and endogeic species [80]. Hence, the probability that larger earthworm clusters were sampled increased in larger habitats.

The observed significant effect of elevation was consistent with results of other studies carried out in mountain habitats, which showed that the diversity of earthworms varied largely in its dependence on altitude [81,82,83]. These studies, however, quantified the influence of the elevation across very long gradients. Nevertheless, our study showed that even a relatively short altitudinal range, from 750 to 1000 m a.s.l., may significantly affect earthworm community composition.

### 4.2. The Effects of Artificial Drainage (Hydrologic Changes and Local Topography) on the Composition and Distribution Pattern of Earthworm Traits—A Functional Approach

Studies based on the analysis of community-weighted mean trait scores within earthworm communities revealed a link between variations in trait values and dynamics of the hydrologic regime [5,7]. Our results supported this conclusion and indicated that the alteration of the natural mountain fen soils by artificial drainage has resulted in community composition changes as summarized by the changes in community mean trait values and led to the replacement of earthworms with different trait values across the examined HC.

The negative, positive and neutral effect achieved in distribution patterns of the community-weighted trait mean scores (Tm) provide important insights suggesting that the earthworm response may be resistant without variation or highly sensitive, showing a change across the examined environmental gradient. However, our findings implied that traits related to HC changes in mountain fens are different from those involved in the response to periodic flooding as a stressor [7] or those across a perturbation gradient in a restored river floodplain [5], among which the high reproduction and diapause or hygrochorous dispersal traits were coping with various intensities of floods and droughts [5,7]. In mountain fens, nested ANOVA results showed that decreasing exposure to inundation induced the shift mainly within the biological and ecological classes of traits, and that this affected such trait values as maximum body length, habitat width, hygrophily tolerance and tolerance to low pH conditions. However, changes in the value of biological traits, such as the presence or absence of inactive stages, were mainly influenced by site-specific conditions nested within the different HC. In consequence, earthworm community composition in the degraded conditions of mountain fens was characterized by increased density of eurytopic worms with longer bodies, preferring less water-saturated and more acidic soils. On the basis of these results, we highlighted that abiotic-based environmental filtering was the main process sorting earthworm traits in the disturbed HC. These results suggested that the environmental filtering (sensu Kraft et al. [84]) and abiotic tolerance of species were the main mechanisms shaping the earthworm trait composition. They played a crucial role in the earthworm community assembly in the degraded mountain fen soils caused by groundwater withdrawal.

The composition of earthworm functional traits related to vertical distribution, and defining various burrowing behaviors across the examined HC were also markedly different in comparison to those occurring in more stressed systems of the river floodplains [5]. According to nested (hierarchical) ANOVA results, the responses of traits representing different eco-morphological groups of earthworms to hydrologic changes were less sensitive and site-specific in mountain fens. Earthworm communities in mountain fens, contrary to those observed in river floodplains, were devoid of anecic subsoil dwellers, and the pattern of the other traits related to vertical distribution and burrowing behavior were consistent across the examined gradient. The observed lack of anecic species and the neutral response of the epigeic, epi-endogeic and endogeic mean trait values to HC changes demonstrates that the thickness of superficial deposits (in our study, maximum 150 cm, after Nicia, personal communication) has an effect on the earthworm community composition at the wetland sites. However, such results also pointed to different components of ecological and landscape memory [85] influencing population stability and resilience in wetland habitat patches within a river floodplain and in the mountain ecosystems, caused by various links between wetland patches and their hydro-geomorphological template [86]. The observed contradiction between a set of earthworm traits reacting to changes in intensities of inundation in a river floodplain to those responding to hydrologic changes in groundwater-fed fens in the mountain landscape support the annotation by Lindo [87] that “traits involved in the response of a species to one environmental stressor may not be relevant in the response to another”. It is also consistent with Plum’s thesis that differences between hydrologic regime dynamics within wetland habitat patches in a river floodplain and those appearing in mountain fens may be of importance to the soil invertebrate response [12].

In the current study, variation partitioning identified that, as in the case of analysis based on species composition, HC and LT had stronger effects on trait variation than the sum of the two separate effects. However, contrary to the analysis based on the community composition, the pure effect of time (season + year) was significant and explained the two-fold greater variation of the earthworm traits in relation to taxonomic composition. This result leads to the conclusion that the spatial and temporal heterogeneity in the physicochemical environment of mountain fens is one of the dimensions on which characteristic species traits can be sorted. Our finding supports the Townsed and Hildrew hypothesis of species traits relation to habitat template [88]. The forward selection procedure supported these findings and discriminated the semi-natural conditions, then elevation, patch size and time, among the different hydrologic disturbance levels as dimensions of the template on which earthworm traits achieved the highest variation and were filtered from a pool of potential colonists of mountain fens. Further, pRDA and GLM analysis results also confirmed that semi-natural conditions create an intermediate phase of disturbances, which had the highest diversifying effect on trait values and allowed the co-existence of species with contrasting habitat requirements. The observed peak of trait value variation under semi-natural HC emphasizes the role of transient dynamics in disturbance regimes [89] and also provides an explanation of the convergence pattern of their responses to environmental change. Our results showed that artificial drainage selected a set of earthworm functional traits of which the type of prostomium, habitat width, hydrophilicity tolerance and low pH tolerance had significant correlations with degraded conditions. This allowed us to determine that abiotic factors, such as the acidity and moisture of soil, were the main environmental filters structuring the earthworm community under degraded conditions and led to the co-occurrence of eurytopic earthworm species with tanylobic prostomiumwhich were more resistant to drought and to highly acidified conditions. The type of prostomium is a trait related to the diet and feeding behavior of earthworms. Tanylobic prostomium represents the most maneuverable trait and enables worms to manipulate plant debris. The species with tanylobic prostomium are not restricted to one ecological group of earthworms, but they mainly include epigeic and anecic earthworms. Their prevalence in degraded sites could reflect both harsher conditions in the mineral soil layer (lower soil moisture) or lower content and availability of humified and fine particulate organic matter in deeper soil horizons for engogeic and amphibious worms with other prostomium types (zygolibic and epilobic) feeding on it. Thus, the documented variation in the community-weighted mean trait values among natural, semi-natural and degraded conditions supported the statement by Decaëns et al. [90] that within the extreme stages of disturbance, the ecological coherence is higher than under intermediate conditions, and that convergence in traits associated with earthworm species responds to environmental constraints.

## 5. Conclusions

Hydrologic regime changes caused by artificial drainage represent a hydrological disturbance mechanism and an environmental stress gradient creating and sustaining species and trait variation of earthworm communities in mountain fens. This disturbance mechanism had a neutral effect on taxonomically based diversity indices, although species-specific positive and negative responses were observed. There was a negative effect of disturbances on the total density of earthworms, which was reduced almost three times under drained compared with natural conditions. At the functional level, the trait responses to the HC changes might be either sensitive or resistant, showing no changes along the examined environmental gradient. Traits reflecting significant variation along the examined hydrologic gradient changes, such as hydrophilicity and low pH tolerance, support the importance of abiotic-based environmental filtering of earthworms under drained conditions and indicate the role of spatial and temporal heterogeneity in the physicochemical environment as one of the dimensions on which characteristic species traits can be sorted. The groundwater decrease led to the co-occurrence of eurytopic, surface browsing, and drought- and acid-resistant earthworm species in the drained mountain fens. The variety peak of the analyzed earthworm functional traits within semi-natural HC emphasizes the role of the transient dynamics in altered disturbance regimes [89] and demonstrates linkages with the severity of hydrologic disturbances across a hydrologic gradient.

Our results based on the comparative analysis of taxonomic and community-weighted mean trait score composition of earthworm communities are consistent with the concept of Mouillot et al. [20] that a trait-based approach can better quantify the impacts of disturbance on ecological communities than the taxonomic approach. Nevertheless, in the various HC in the mountain fens, the functional traits approach had only slightly more predictive power than the taxonomic one, although it showed better which processes were responsible for earthworm species sorting. Our data also provided evidence that environmental change within wetland habitat patches, which evolved under various hydrologic–edaphic conditions and dynamics of the hydrologic regime, sorted different sets of earthworm traits involved in the response. They also supported the concept by Lindo et al. [87] that different kinds of environmental stressors (flooding versus perennially saturation of soils) could have various impacts on the traits involved in the response and showed that different earthworm traits were responsive to environmental changes in the hydrologic regime dynamic within wetland habitat patches in river floodplains [5] and those appearing in mountain fens with perennially saturated soils, albeit without obvious flooding. Our results for the first time showed that artificial drainage had an impact on selected earthworm functional traits, of which the type of prostomium, habitat width, hydrophilicity tolerance and low pH tolerance had significant correlations with degraded conditions. This allowed us to identify abiotic factors, such as the acidity and moisture of soil, as the main environmental filters structuring the earthworm community under degraded conditions. These two environmental factors led to co-occurrence of eurytopic earthworm species with tanylobic prostomium, which are more resistant to drought and to highly acidified conditions.

## Figures and Tables

**Figure 1 biology-12-00482-f001:**
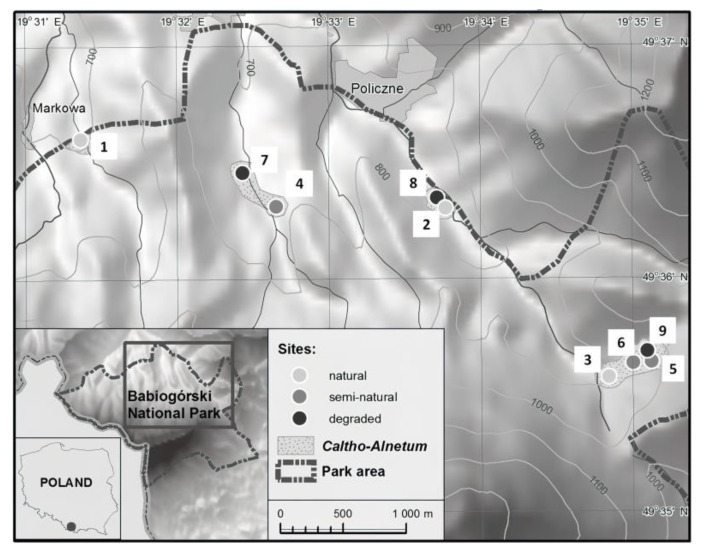
Location of the Babiogórski National Park in the Polish Flysh Outer Carpathians and location of the studied mountain fen *Caltho-Alnetum* sites with different degrees of artificial drainage (hydrologic condition, HC); 1–9 sites numbering. Acknowledgement: This figure was published in European Journal of Soil Biology, 68, Maria Sterzyńska, Karel Tajovský, Paweł Nicia, Contrasting responses of millipedes and terrestrial isopods to hydrologic regime changes in forested montane wetlands, 33–41, Copyright Elsevier (2015).

**Figure 2 biology-12-00482-f002:**
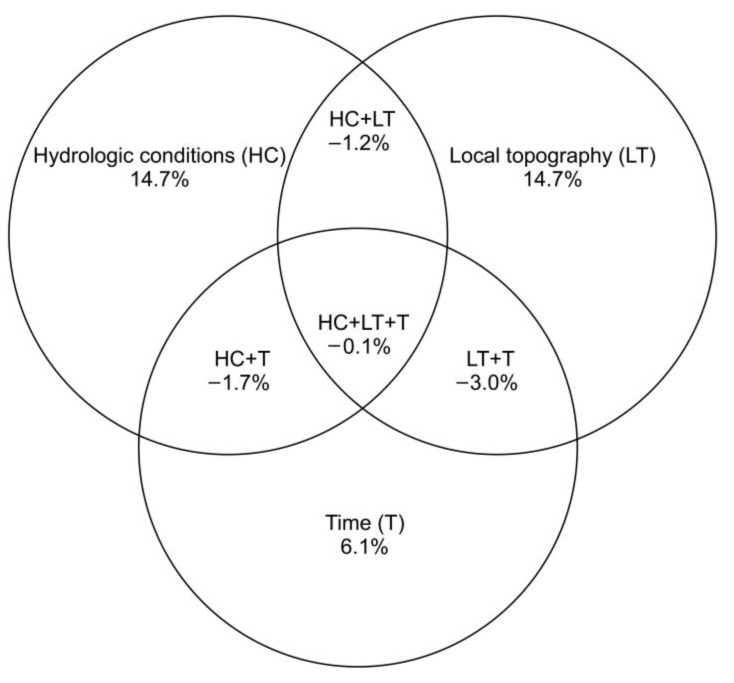
Venn diagram showing overlap of hydrologic changes (HC), local topography (LT) and time (T) based on earthworm community composition in mountain fens. Community composition fit with earthworm species abundance data; variation partitioning based on pRDA analysis.

**Figure 3 biology-12-00482-f003:**
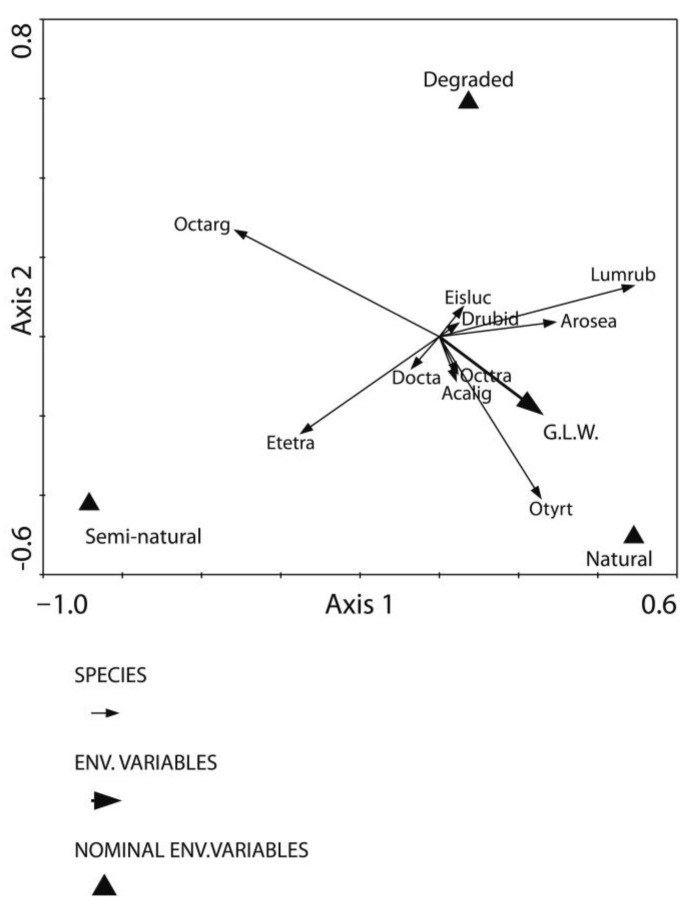
Ordination of earthworm community constrained by hydrologic conditions (HC) and local topography (LT) of the mountain fens of *Caltho-Alnetum*. pRDA model calculated with log(x + 1)-transformed species data, interspecies correlation scaling, species scores divided by SD and centered by species, not standardized by sample; time (years of study and season) and local topography used as covariates.

**Figure 4 biology-12-00482-f004:**
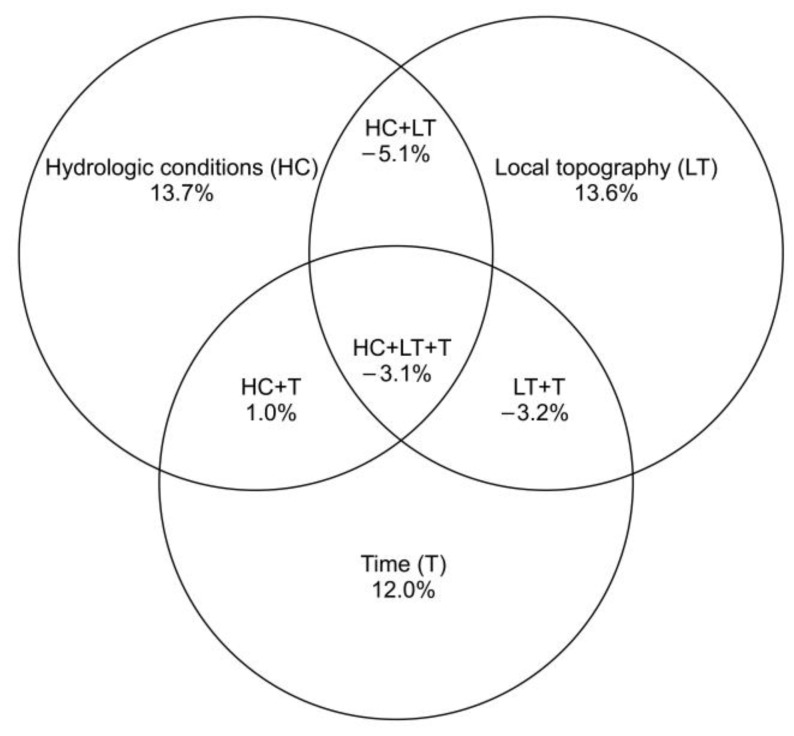
Venn diagram showing overlap of hydrologic changes (HC), local topography (LT) and time (T) on earthworm community composition in mountain fens. Community composition fit with earthworm traits; variation partitioning based on pRDA analysis.

**Figure 5 biology-12-00482-f005:**
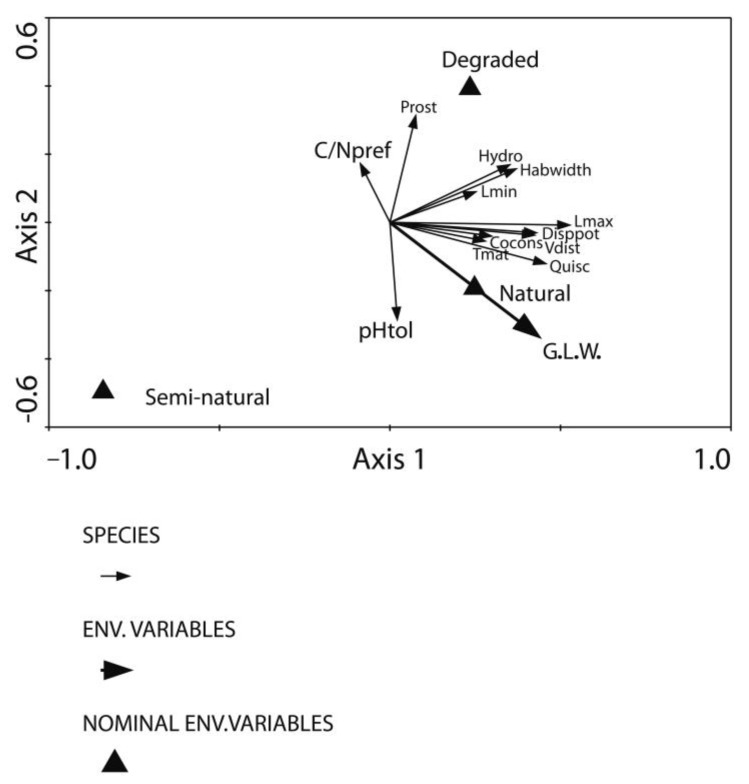
Ordination of Lumbricidae traits constrained by hydrologic conditions (HC) in mountain fens of *Caltho-Alnetum*. pRDA model calculated with log_10_(x + 1)-transformed species data (= traits), interspecies correlation scaling, species scores divided by SD and centered by species, not standardized by sample; time (years of study and season) and local topography used as covariates.

**Table 1 biology-12-00482-t001:** Characteristics of the mountain fen sites (1–9) of *Caltho-Alnetum* in different hydrologic conditions (HC). Soil type classification was based on the World Reference Base for Soil Resources [48].

Hydrologic Conditions	Size (m^2^)	Slope (°)	Elevation (m a.s.l.)	Soil Type
Natural				
1	1775	5	914	Otni
2	750	2	772	POtni
3	546	5	703	POtni
Semi-natural				
4	255	8	964	POtni
5	945	7	950	POtni
6	897	3	760	POtni
Degraded				
7	1284	3	962	Mtni
8	12,903	3	771	Mtni
9	739	6	740	Ah

**Table 2 biology-12-00482-t002:** Attribute classes, types of data and codes of earthworm functional traits.

Traits	Data Type	Attribute Class	Code
Biological traits:			
Length min/max at maturity stage (mm)	ordinal	1—<50 mm	Lmin/max
		2—50–90 mm	
		3—>90 mm	
Prostomium	ordinal	1—epilobic open	Prost
		2—epilobic closed	
		3—tanylobic	
Dispersal potential	ordinal	1—high	Dispot
		2—intermediate	
		3—low	
Quiescence	binary	1—no	Quisc
		2—yes	
Performance traits:			
Time to maturity (weeks)	binary	1—<52	Tmat
		2—>52	
Cocoons (number/year)	binary	1—<36	Cocons
		2—>36	
Ecological traits:			
Vertical distribution	ordinal	1—epigeic	Vdist
		2—epiendogeic	
		3—endogeic	
Habitat width	binary	1—steno	Habwidth
		2—eury	
Hygrophily	ordinal	1—high	Hydro
		2—intermediate	
		3—low	
C/N soil preference	ordinal	1—low	C/Npref
		2—eury	
		3—high	
Low pH tolerance	binary	1—tolerant	pHtol
		2—euryvalent	

All data types are ordinal/binary; functional classes are according to [16].

**Table 3 biology-12-00482-t003:** Soil characteristics of mountain fen sites of *Caltho-Alnetum* in different hydrologic conditions (HC). Values presented in the table are means across replicates of each site (n = 3) with standard deviation (±SD). Mg^2+^—Mg^2+^ cation concentration; EC—electrical conductivity; TEB—total exchangeable base cations; Hh—hydrolytic acidity; CEC—cation exchange capacity; BS—base saturation; G.W.L.—ground water level; (n = 6).

Soil Properties	Natural		Semi-Natural		Degraded	
	Mean	SD	Mean	SD	Mean	SD
Mg^2+^ * (μmol kg^−1^ soil)	86.11	8.68	52.78	6.01	21.06	16.07
EC *	0.19	0.08	0.12	0.04	0.06	0.02
TEB * (μmol kg^−1^ soil)	919.28	268.62	598.02	92.77	125.53	132.46
Hh(=H^+^) * (μmol kg^−1^ soil)	6.78	2.03	7.18	0.66	137.21	54.48
CEC	926.06	266.74	605.20	92.84	262.74	112.73
BS * (%)	99.13	0.55	98.79	0.19	72.34	5.33
pHH2O	6.50	0.45	5.28	0.29	4.72	0.66
C/N	22.68	3.64	19.08	2.32	20.09	2.69
G.W.L. * (cm)	3.53	0.78	14.39	3.08	22.17	13.43

* *p* < 0.05 level of statistical significance tested by Kruskal–Wallis one-way ANOVA.

**Table 4 biology-12-00482-t004:** Effect of hydrologic conditions (HC) and fen sites on the species of Lumbricidae and community metrics: species mean abundance and total abundance A (ind. m^2^), species richness (S), Shannon diversity index (H’) and Pielou evenness index (J’). Two-way nested ANOVA model was calculated with log_10_(x + 1)-transformed data except for (H’) and (J’) indices (HC–d.f. = 2; sites d.f. = 6). F—value of partial F test statistic; K—value of Kruskal Wallis test statistic; letters in row—indicate significant differences among HC tested post-hoc by the Tukey HSD test. Values presented in the table are not transformed. Means across replicates of each site (n = 3) with standard deviation (± SD).

Species	Code	Natural	Semi-Natural	Degraded	HC		Site	
					F/K	*p*	F	*p*
*Aporrectodea caliginosa* (Savigny, 1826)	Acalig	8.00 ± 12.31	8.59 ± 15.21	8.30 ± 19.89	0.16	0.854	1.52	0.193
*Aporrectodea rosea* (Savigny, 1826)	Arosea	5.63 ± 8.07	2.07 ± 3.24	5.33 ± 12.54	0.53	0.595	1.42	0.228
*Dendrobaena octaedra* (Savigny, 1826)	Docta	58.67 ± 35.0	35.56 ± 26.06	53.33 ± 43.79	3.85	0.146		
*Dendrodrilus rubidus* (Savigny, 1826)	Drubid	3.85 ± 7.50	4.15 ± 8.29	2.96 ± 4.92	0.002	0.998	1.04	0.412
*Eisenia lucens* (Waga, 1875)	Eisluc	4.15 ± 7.21	0.59 ± 1.72	2.67 ± 5.56	2.43	0.297		
*Eiseniella tetraedra* (Savigny, 1826)	Etetra	124.44 ± 191.64 b	69.33 ± 43.13 c	3.85 ± 7.27 a	19.61	0.001 *		
*Lumbricus rubellus* Hoffmeister, 1845	Lumru	3.56 ± 6.84 ab	1.85 ± 3.90 b	5.04 ± 7.42 a	3.23	0.049 *	3.54	0.006 *
*Octodrilus argoviensis* (Bretscher, 1899)	Octarg	0.30 ± 1.26 a	14.52 ± 19.43 b	0.00 ± 0.00 a	20.95	0.000 *		
*Octodrilus transpadanus* (Rosa, 1884)	Octtra	1.19 ± 3.90	0.30 ± 1.26	0.00 ± 0.00	1.23	0.334	0.65	0.692
*Octolasion tyrtaeum* (Savigny, 1826)	Otyrt	40.59 ± 58.20 b	13.33 ± 17.76 ab	7.11 ± 11.57 a	5.10	0.010 *	6.53	0.000 *
A (total)		250.37 ± 244.53 b	149.63 ± 80.71 ab	88.59 ± 49.06 a	6.24	0.004 *	6.94	0.000 *
S		4.22 ± 1.63	4.44 ± 1.20	3.56 ± 1.15	2.17	0.125	0.90	0.501
H’		1.07 ± 0.40	1.17 ± 0.25	0.84 ± 0.36	4.55	0.015	1.50	0.200
J’		0.77 ± 0.14	0.81 ± 0.09	0.68 ± 0.20	4.07	0.131		

Significant *p* value is shown in *.

**Table 5 biology-12-00482-t005:** Ranking environmental variables by importance according to marginal and conditional effects on the Lumbricidae community taxonomic composition variability in the mountain fen *Caltho-Alnetum* sites by automatic forward selection in RDA. RDA model calculated with log_10_(x + 1)-transformed species data. λ1 = fit = eigenvalue with only one variable; λa = additional fit = increase in eigenvalue; cum %—explained cumulative % of eigenvalues; F—value of the F-ratio statistic; *p*—significance level of the conditional effect tested by unrestricted Monte Carlo permutation test under the reduced model with 499 permutations; G.W.L.—groundwater level.

Marginal Effect (Forward: Step 1)			Conditional Effect (Forward: Continued)		
Variable	λ1	λa	Cum %	F	*p*
Size	0.05	0.05	5	3.01	0.008 *
Semi-natural	0.05	0.04	9	2.27	0.058
Elevation	0.03	0.09	18	5.43	0.002 *
Slope	0.05	0.03	21	1.81	0.094
Time (year 2010)	0.02	0.03	24	1.55	0.116
Natural	0.03	0.02	26	1.40	0.198
G.W.L	0.03	0.04	30	2.74	0.018
Time (year 2011)	0.02	0.02	32	1.00	0.406
Time (spring)	0.01	0.01	33	0.63	0.710
Degraded	0.05				
Time (year 2012)	0.02				
Time (spring)	0.01				
Time (autumn)	0.01				

Significant *p* values are shown in *.

**Table 6 biology-12-00482-t006:** Effect of hydrologic conditions (HC) and fen sites on the Lumbricidae traits. Two-way nested ANOVA model calculated with log_10_(x + 1)-transformed data. F—value of partial F test statistic, K—value of Kruskal–Wallis test statistic; letters in row—indicate significant differences among HC tested post-hoc by the Tukey HSD test. Traits coding as in Table 2.

Traits	Natural	Semi-Natural	Degraded	HC		Site	
				F/K	*p*	F	*p*
Biological traits:							
Lmin (mm)	2.04 ± 0.03	2.03 ± 0.04	2.06 ± 0.06	1.60	0.219	1.10	0.357
Lmax (mm)	2.36 ± 0.03	2.34 ± 0.04	2.37 ± 0.05	6.32	0.042 *		
Prostomium	2.07 ± 0.08	2.10 ± 0.07	2.10 ± 0.10	0.71	0.499	2.23	0.056
Dispersal potential	2.16 ± 0.06	2.11 ± 0.06	2.14 ± 0.13	4.26	0.119		
Quiescence	2.17 ± 0.08	2.11 ± 0.11	2.17 ± 0.14	1.46	0.244	2.56	0.032 *
Performance traits:							
Time to maturity	2.17 ± 0.09	2.17 ± 0.09	2.17 ± 0.14	0.01	0.990	2.09	0.073
Cocoons (n/year)	2.19 ± 0.07	2.17 ± 0.09	2.18 ± 0.13	0.77	0.681		
Ecological traits:							
Vertical distribution	2.18 ± 0.11	2.15 ± 0.10	2.18 ± 0.14	0.60	0.554	2.37	0.045 *
Habitat width	2.37 ± 0.11 b	2.28 ± 0.08 a	2.44 ± 0.08 c	24.83	0.000 *	7.00	0.000 *
Hygrophily	2.26 ± 0.09 b	2.19 ± 0.07 a	2.31 ± 0.07 c	19.18	0.000 *	6.43	0.000 *
C/N soil preference	2.38 ± 0.06	2.36 ± 0.06	2.40 ± 0.08	1.62	0.209	2.83	0.020 *
Low pH tolerance	2.27 ± 0.15	2.35 ± 0.05	2.20 ± 0.14	7.80	0.02 *		

Significant *p* value is shown by *.

**Table 7 biology-12-00482-t007:** Ranking environmental variables by importance according to marginal and conditional effects based on the Lumbricidae traits pattern in the mountain fen *Caltho-Alnetum* sites by automatic forward selection in RDA. RDA model calculated with log_10_(x + 1)-transformed species traits data. λ1 = fit = eigenvalue with only one variable; λa = additional fit = increase in eigenvalue; cum %—explained cumulative % of eigenvalues; F—value of the F-ratio statistic; *p*—significance level of the conditional effect tested by unrestricted Monte Carlo permutation test under the reduced model with 499 permutations; G.L.W.—groundwater level.

Marginal Effect (Forward: Step 1)			Conditional Effect (Forward: Continued)		
Variable	λ1	λa	cum %	F	*p*
Semi-natural	0.07	0.07	7	3.92	0.020 *
Elevation	0.05	0.10	17	6.47	0.012 *
Size	0.07	0.05	22	3.20	0.038 *
Time (year 2010)	0.05	0.05	27	3.07	0.036 *
Time (spring)	0.03	0.03	30	1.92	0.152
G.W.L.	0.02	0.03	33	2.15	0.086
Natural	0.01	0.03	36	2.01	0.124
Slope	0.05	−0.00	36	0.58	0.616
Time (year 2011)	0.04	0.01	37	0.47	0.678
Degraded	0.04				
Time (autumn)	0.03				
Time (year 2012)	0.01				

Significant *p* values is shown by *.

**Table 8 biology-12-00482-t008:** Parameters and fitting values of the generalized linear models (GLMs) for mean functional traits of Lumbricidae. Results constrained by hydrologic conditions (HC); time (T) and local topography (LT) used as covariates. Regression coefficient (Coef. = slope), SD—standard deviation, t—value statistic to test if the coefficient is different from zero, DFs—residual deviance, F—value statistic to test the fit of the model after introducing each variable (df for null model = 53, for Ax1 and Ax2 = 52 for all the traits). Trait codes as in Table 2.

	Coef. (±SD)	t	DFs.	F	*p*		Coef. (±SD)	t	DFs	F	*p*
Biological traits											
Lmin											
null			71.80			null			71.80		
intercept	4.695 (0.013)	360.716				intercept	4.695 (0.013)	360.716			
Ax1			71.80		ns	Ax2			71.80		ns
Lmax											
null			117.81			null			117.81		
intercept	5.428 (0.009)	601.421				intercept	5.429 (0.009)	602.353			
Ax1	0.051 (0.009)	5.686	85.51	19.55	<0.001 *	Ax2			117.81		ns
Prost											
null			287.73			null			287.73		
intercept	4.821 (0.012)	3.94.724				intercept	4.820 (0.012)	393.649			
Ax1	-	-	287.73		ns	Ax2	0.067 (0.012)	5.508	257.46	5.87	0.020 *
Disppot											
null			363.32			null			363.32		
intercept	4.938 (0.012)	514.0.84				intercept	4.943 (0.011)	429.858			
Ax1	0.101 (0.011)	7.254	285.33	14.19	<0.001 *	Ax2			363.32		ns
Quisc											
null			560.97			null			560.97		
intercept	4.970 (0.011)	436.517				intercept	4.977 (0.011)	440.399			
Ax1	0.121 (0.011)	10.745	445.96	13.20	<0.001 *	Ax2			560.97		ns
Performance traits											
Tmat											
null			505.68			null			505.68		
intercept	5.015 (0.011)	451.913				intercept	5.018 (0.011)	453.465			
Ax1	0.075 (0.011)	6.818	459.40	5.22	0.026 *	Ax2			505.68		ns
Cocoons											
null			425.47			null			425.47		
intercept	5.044 (0.010)	461.124				intercept	5.047 (0.011)	462.549			
Ax1	0.071 (0.010)	6.571	382.28	5.83	0.019 *	Ax2			425.47		ns
Ecological preference											
Vdist											
null			580.91			null			580.91		
intercept	5.013 (0.011)	450.267				intercept	5.019 (0.011)	453.895			
Ax1	0.115 (0.011)	10.457	472.01	11.75	<0.001 *	Ax2			580.91		ns
Habwidth											
null			757.28			null			757.28		
intercept	5.466 (0.009)	616.719				intercept	5.373 (0.013)	411.439			
Ax1	0.091 (0.009)	10.416	649.05	9.26	0.007 *	Ax2	0.023 (0.009)	2.674	629.29	5.49	0.007 *
Hydro											
null			429.54			null			429.54		
intercept	5.201 (0.010)	514.084				intercept	5.122 (0.015)	344.297			
Ax1	0.073 (0.010)	7.254	376.92	7.53	0.008 *	Ax2	0.023 (0.009)	2.355	357.54	5.34	0.008 *
C/Npref											
null			284.91			null			284.91		
intercept	5.486 (0.008)	626.358				intercept	5.486 (0.009)	626.358			
Ax1			284.91		ns	Ax2			284.91		ns
pHtot											
null			927.18			null			927.18		
intercept	5.278 (0.010)	543.321				intercept	5.363 (0.014)	386.442			
Ax1			927.18		ns	Ax2	−0.081 (0.010)	−7.884	760.58		0.006 *

Significant *p* value is shown in *.

## Data Availability

The authors confirm that the data supporting the findings of this study are available within the article [and/or] its Appendix A.

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
