# Peer review of "Effects of Hydrologic Regime Changes on a Taxonomic and Functional Trait Structure of Earthworm Communities in Mountain Wetlands"

_biology, 2023, doi:10.3390/biology12030482_

Round 1
Reviewer 1 Report
Authors investigated effects of human-induced changes in mountane wetlands hydrology on species composition and functional traits of earthworm communities. Research was conducted in Babiogorski National Park (Poland). Sampling included three types of wetlands, representing different degree of human-induced impacts, such as: (1) degraded (drained), (2) semi-natural (drained, but with not active drainage system) and (3) natural (without artificial drainage). Three sampling sites were established within each type of wetland and sampled during three consecutive years, twice (in spring and fall) each year. Such sample size seems to be a bit small to me; still results of statistical analysis prove to be sufficient. Analysis of data seems to be well executed and results are generally well presented.
However, I am truly sorry to say, but the manuscript itself is not really well written and some parts need quite substantial revision. My biggest criticism concerns language style and grammar, and the manuscript structure, particularly in case of Introduction and Discussion. In addition to a number of obvious grammar mistakes, many sentences are very long and complex; they also contain digressions which make them difficult to follow and increases number of mistakes in sentences grammar and structure. The text will greatly benefit if author brake down such sentences into simple once with clear focus. Some paragraphs also face the same problems; they are long and often contain several different topics. This is particularly refers to the Introduction, Discussion and Conclusions, with the last one composted even of only a single long paragraph. Breaking down some of these very long paragraphs would certainly make the text easier to follow.
I have also serious concerns regarding usage of essential terminology. In particular, authors usage and understanding of the term "disturbance". Although the very first sentence of introduction starts with that word (implying that the work focuses on investigating the effects of disturbances) it is not clear which factor is considered by authors as "disturbance" in case of the investigated montane wetlands. Only Conclusions clearly define that, such information should be clearly given already in Introduction. The other problem is that authors define disturbance as "hydrologic regime changes caused by artificial drainage" (see the first sentence of Conclusions). However, as limited resources (such as water in that case) are generally not considered as "disturbance" but as "stress" (e.g., Grime 1979) considering long lasting (permanent) change in hydrology cannot be considered as disturbance. While this does not invalidate the results of this work, it makes large part of Introduction irrelevant.
I have also provided numerous more detailed comments and suggestions directly in the manuscript. However, regarding language corrections or manuscript structure I have only marked some parts that are in need of revision.
------------------------------------------------------------------------------

Author Response
Rev. 1
Authors investigated effects of human-induced changes in mountane wetlands hydrology on species composition and functional traits of earthworm communities. Research was conducted in Babiogorski National Park (Poland). Sampling included three types of wetlands, representing different degree of human-induced impacts, such as: (1) degraded (drained), (2) semi-natural (drained, but with not active drainage system) and (3) natural (without artificial drainage). Three sampling sites were established within each type of wetland and sampled during three consecutive years, twice (in spring and fall) each year. Such sample size seems to be a bit small to me; still results of statistical analysis prove to be sufficient. Analysis of data seems to be well executed and results are generally well presented.
Authors’ Response:
Thank you for the positive opinion about our study.
However, I am truly sorry to say, but the manuscript itself is not really well written and some parts need quite substantial revision. My biggest criticism concerns language style and grammar, and the manuscript structure, particularly in case of Introduction and Discussion. In addition to a number of obvious grammar mistakes, many sentences are very long and complex; they also contain digressions which make them difficult to follow and increases number of mistakes in sentences grammar and structure. The text will greatly benefit if author brake down such sentences into simple once with clear focus. Some paragraphs also face the same problems; they are long and often contain several different topics. This is particularly refers to the Introduction, Discussion and Conclusions, with the last one composted even of only a single long paragraph. Breaking down some of these very long paragraphs would certainly make the text easier to follow.
Authors’ Response:
Thank you, we corrected the text and sent it back for English verification
Rev. 1
I have also serious concerns regarding usage of essential terminology. In particular, authors usage and understanding of the term "disturbance". Although the very first sentence of introduction starts with that word (implying that the work focuses on investigating the effects of disturbances) it is not clear which factor is considered by authors as "disturbance" in case of the investigated montane wetlands. Only Conclusions clearly define that, such information should be clearly given already in Introduction. The other problem is that authors define disturbance as "hydrologic regime changes caused by artificial drainage" (see the first sentence of Conclusions). However, as limited resources (such as water in that case) are generally not considered as "disturbance" but as "stress" (e.g., Grime 1979) considering long lasting (permanent) change in hydrology cannot be considered as disturbance. While this does not invalidate the results of this work, it makes large part of Introduction irrelevant.
Authors’ Response
We considered a disturbance as "any event that is relatively discrete in time and space that disrupts the structure of an ecosystem, community, or population and changes resources availability or the physical environment" (White and Pickett, 1985). Although hydrologic disturbances may be considered discrete events, they are not always punctuated by a clear beginning or end, particularly in cases where external drivers are needed for the hydrologic impacts to be expressed. This concept allows us to consider the wetland hydrological regime as an ecological disturbance associated with natural, semi-natural and degraded hydrological conditions (hydrological disturbance caused by artificial drainage = human activity). According to the disturbance synthesis (Graham et al. 2021 Toward a generalizable framework of disturbance ecology …), "...disturbances are often inferred to be synonymous with pulse events, perturbations, threats and/or stressors...". Hydrological regime and/or hydrological conditions in wetlands (pulses of inundation, water regime as natural environmental factors) lead to the process of paludification - the accumulation of organic matter over time and the development of hydric soils characterized by low microbial activity and nutrient availability; drainage of wetlands through changes in hydrological regime and/or conditions leads to ecological change, e.g. an increase in the rate of decomposition, microbial activity and nutrient availability and change ecosystem trajectory. We agree that disturbance interacts with environmental stressors, and in wetlands, e.g. inundation, water resources and nutrient availability are environmental stressors. We related, however, our study to the various hydrologic regime (wetland hydrological conditions) as disturbance drivers.
In the introduction, we improved the main objectives, showing what type of disturbances was studied.
Rev. 1
I have also provided numerous more detailed comments and suggestions directly in the manuscript. However, regarding language corrections or manuscript structure I have only marked some parts that are in need of revision.
Authors’ Response
Following reviewer’s suggestion we have corrected the specified parts of the text. We have provided answers to the comments in the pdf file.

Reviewer 2 Report
The study is relevant. The potential of soil invertebrates as biological indicators of anthropogenic effects on natural ecosystems is known.
The objectives of the present study were 1) to quantify the response of earth-worm community composition and community-weighted functional traits of species to hydrologic changes, 2) to evaluate which functional traits of earth-worms contribute to the prediction of hydrologic change and 3) to assess the relationship between patterns of earthworm species, community-weighted means of earthworm traits and environmental variables.
The authors suggested that the drainage-related changes of mountain wetlands soils have an effect on community composition and functional structure of soil biota.
Through field study has been demonstrated for the first time that hydrological disturbances affected the functional and taxonomic composition of soil biota represent by earthworms.
Results obtained by the authors based on the comparative analysis of taxonomic and community-weighted mean trait score composition of earthworm communities confirm the concept that the trait-based approach can better quantify the impacts of disturbance on ecological communities than the taxonomic approach.
1. The research topic is sufficiently substantiated, the methods are presented quite fully. 2. Conclusions correspond to the obtained results. They answer the question. 3. Mandatory editing is not required. Possible technical edition at the discretion of the authors. 4. Tables and figures are very informative. Figures could be done in color.
The results obtained are of both theoretical and practical importance for diagnosing disturbances in wetlands. The submitted manuscript may be published in the journal Biology.
Author Response
Rev. 2
The study is relevant. The potential of soil invertebrates as biological indicators of anthropogenic effects on natural ecosystems is known.
The objectives of the present study were 1) to quantify the response of earthworm community composition and community-weighted functional traits of species to hydrologic changes, 2) to evaluate which functional traits of earth-worms contribute to the prediction of hydrologic change and 3) to assess the relationship between patterns of earthworm species, community-weighted means of earthworm traits and environmental variables.
The authors suggested that the drainage-related changes of mountain wetlands soils have an effect on community composition and functional structure of soil biota.
Through field study has been demonstrated for the first time that hydrological disturbances affected the functional and taxonomic composition of soil biota represent by earthworms.
Results obtained by the authors based on the comparative analysis of taxonomic and community-weighted mean trait score composition of earthworm communities confirm the concept that the trait-based approach can better quantify the impacts of disturbance on ecological communities than the taxonomic approach.
- The research topic is sufficiently substantiated, the methods are presented quite fully. 2. Conclusions correspond to the obtained results. They answer the question. 3. Mandatory editing is not required. Possible technical edition at the discretion of the authors. 4. Tables and figures are very informative. Figures could be done in color.
The results obtained are of both theoretical and practical importance for diagnosing disturbances in wetlands. The submitted manuscript may be published in the journal Biology.
Authors’ Response:
Thank you for the positive opinion about our study.
Reviewer 3 Report
This will be a very nice contribution to our knowledge of the Functional traits and structure of earthworm communities in mountain wetlands. This is an interesting study and the authors have collected a unique dataset using a cutting-edge methodology. The paper is generally well-written and structured. Both the title, the simple summary, and the abstract are very well constructed to be informative for continuing the reading. I especially appreciated the introduction, which was very informative and provided an up-to-date review of the main topic of the manuscript (MS). I do believe that the manuscript needs some minor additional work before it can be published, but I have made a few comments; suggestions for edits, but I believe they can be easily handled or refuted by the authors. Some general comments are included below.
I will look forward to seeing the published paper and congratulate the authors on the work done here.
Sincerely,
____________________
In my opinion, table headings or figure footnotes would be better framed in the text if a font size of 9 could be used (I am not sure, and the authors do not have to follow my suggestion).
Regarding the Results section, I noticed the absence of the taxonomic authorship of the species in Table 4. You should add them to the text the first time you cite a species or as an idea, in order not to modify the MS, the authors could prepare a supplementary table with the authorship of the specie, and make some links to the table the first time you cite the species.
Maybe my next comment is more of a doubt, but it would look more visual if it were possible to unify some results tables. Do the authors see any possibility of unifying the results tables in some way? As a reader, I would like to see the information all reunited, In my view that will help to follow the MS.
From the rest, I congratulate the authors for such a well-written and designed work. It has been a pleasure to read. I will look forward to seeing the published paper and congratulate the authors on the work done here. The article is almost ready for publication.
Author Response
Rev. 3
This will be a very nice contribution to our knowledge of the Functional traits and structure of earthworm communities in mountain wetlands. This is an interesting study and the authors have collected a unique dataset using a cutting-edge methodology. The paper is generally well-written and structured. Both the title, the simple summary, and the abstract are very well constructed to be informative for continuing the reading. I especially appreciated the introduction, which was very informative and provided an up-to-date review of the main topic of the manuscript (MS). I do believe that the manuscript needs some minor additional work before it can be published, but I have made a few comments; suggestions for edits, but I believe they can be easily handled or refuted by the authors. Some general comments are included below.
I will look forward to seeing the published paper and congratulate the authors on the work done here.
Sincerely,
Authors’ Response:
Thank you for the positive opinion about our study.
Rev. 3
In my opinion, table headings or figure footnotes would be better framed in the text if a font size of 9 could be used (I am not sure, and the authors do not have to follow my suggestion).
Regarding the Results section, I noticed the absence of the taxonomic authorship of the species in Table 4. You should add them to the text the first time you cite a species or as an idea, in order not to modify the MS, the authors could prepare a supplementary table with the authorship of the specie, and make some links to the table the first time you cite the species.
Maybe my next comment is more of a doubt, but it would look more visual if it were possible to unify some results tables. Do the authors see any possibility of unifying the results tables in some way? As a reader, I would like to see the information all reunited, In my view that will help to follow the MS.
Authors’ Response:
We add taxonomic authorship in Table 4, and we add supplementary material
From the rest, I congratulate the authors for such a well-written and designed work. It has been a pleasure to read. I will look forward to seeing the published paper and congratulate the authors on the work done here. The article is almost ready for publication.
Authors’ Response:
Thank you
Reviewer 4 Report
These authors used functional traits to understand the change in earthworm communities after environmental change in fens. This is a very fruitful approach (as judged by studies of other taxa) but has not been frequently used in earthworms.
I have some suggestions for revision of the paper, detailed below. Generally, I think the paper would benefit from editing. There are many aspects of the analysis that don't illuminate much but might distract from more interesting results. Similarly, the writing can be streamlined in many places, and the number of references might be reduced in some places (to just the key references).
Summary and Abstract
I was not sure what the function of the simple summary was. I found that it did not provide as much information about the findings of the study as I was expecting. In particular, I found the last sentence confusing. The abstract was more useful.
Introduction
Well-constructed introduction.
Methods
The design of this study was, to my way of thinking, not nested, although some of the methods and results are phrased that way. In testing for the effect of site, the authors phrase this random factor as being "nested" within hydrologic conditions. I believe that the most proper way to represent this design (and analyze the data) is that the effect of site was examined after blocking for HC (not nested within HC). Perhaps this is what the authors actually did, but this is not clear from the paper.
On the same topic, I am not sure that the effect of site is very important. It seems pretty self-evident that all sites will be a bit different from one another. I did not find the insights gained by that part of the study very interesting. Similarly, the inclusion of year and sample as repeated-measures factors also did not add a lot to the study. The inclusion of site and time factors in these analyses actually served to obscure the more interesting conclusions. I would suggest that the authors consider analyses that do not use those factors as predictive.
I strongly suggest that these authors make their functional trait assignments for the earthworms available, along with the data on which those decisions were made, in the supplemental information for this paper.
In Table 2, I think the authors are using the term "ordinary" where they mean "ordinal."
Here and elsewhere, I think the subscripts are improperly formatted. In particular, I think "log(x10+1)" actually should be log10(x+1).
Results
The family Lumbricidae was specifically mentioned in Table 4. Were Sparganophilus or any other non-lumbricid earthworms found?
On page 10 the paper states "... L. rubellus decreased significantly in degraded fen sites." This is not supported by the data in Table 4.
Generally, I did not find the results of the multivariate analyses very useful. Figures 3 and 5 were really difficult for me to interpret. If the authors want to retain those, they should be clarified.
Some of the significant p-values in Table 6 were not bolded.
I wonder if some of the comparisons between the taxonomic analysis and functional analysis can be made more explicit by placing the comparable data in the same table. For example, could tables 5 and 7 be combined?
Discussion
The part about the loss of alder trees surprised me. I suggest introducing this as a possible correlate of fen drainage in the introduction.
On page 18 the primary author of the paper is referenced using "personal communication." Shouldn't this be "personal observation?"
The authors should speculate about the possible functional significance of the prostomium trend they saw. Why might it be better to have a tanylobic prostomium in degraded soils? If there is no logical reason, then this must simply be reflecting the species differences and not really functional differences.
Conclusion
I suggest breaking this into two paragraphs.
Author Response
Rev. 4
These authors used functional traits to understand the change in earthworm communities after environmental change in fens. This is a very fruitful approach (as judged by studies of other taxa) but has not been frequently used in earthworms.
I have some suggestions for revision of the paper, detailed below. Generally, I think the paper would benefit from editing. There are many aspects of the analysis that don't illuminate much but might distract from more interesting results. Similarly, the writing can be streamlined in many places, and the number of references might be reduced in some places (to just the key references).
Summary and Abstract
I was not sure what the function of the simple summary was. I found that it did not provide as much information about the findings of the study as I was expecting. In particular, I found the last sentence confusing. The abstract was more useful.
Authors’ Response:
We improved “simple summary”
Introduction
Well-constructed introduction.
Authors’ Response:
Thank you
Rev. 4
Methods
The design of this study was, to my way of thinking, not nested, although some of the methods and results are phrased that way. In testing for the effect of site, the authors phrase this random factor as being "nested" within hydrologic conditions. I believe that the most proper way to represent this design (and analyze the data) is that the effect of site was examined after blocking for HC (not nested within HC). Perhaps this is what the authors actually did, but this is not clear from the paper.
Authors’ Response:
We agree, however, we calculated fixed effect model ANOVA
Rev. 4
On the same topic, I am not sure that the effect of site is very important. It seems pretty self-evident that all sites will be a bit different from one another. I did not find the insights gained by that part of the study very interesting. Similarly, the inclusion of year and sample as repeated-measures factors also did not add a lot to the study. The inclusion of site and time factors in these analyses actually served to obscure the more interesting conclusions. I would suggest that the authors consider analyses that do not use those factors as predictive.
I strongly suggest that these authors make their functional trait assignments for the earthworms available, along with the data on which those decisions were made, in the supplemental information for this paper.
Authors’ Response:
Yes, we will add the supplementary materials
In Table 2, I think the authors are using the term "ordinary" where they mean "ordinal."
Authors’ Response:
Done
Rev. 4.
Here and elsewhere, I think the subscripts are improperly formatted. In particular, I think "log(x10+1)" actually should be log10(x+1).
Authors’ Response:
Thank you, we change
Rev. 4.
Results
The family Lumbricidae was specifically mentioned in Table 4. Were Sparganophilus or any other non-lumbricid earthworms found? –
Authors’ Response.
There were no non-lumbricid earthworm found at our sites.
Sparganophilus is native in Nearctic zone (USA). Its occurrence outside of America is limited to few records of S. tamesis in northern and central European countries: England (Benham, 1892; Friend, 1911, 1921 as Helodrilus elongatus, then Sparganophilus elongatus; ÄŒernosvitov, 1945; Jamieson, 1971; Sherlock and Carpenter, 2009); France (Tétry, 1934 as Pelodrilus cuenoti); Switzerland (Lake Lèman in Geneva; Zicsi and Vaucher, 1987; Bouché and Qiu, 1998 as Sparganophilus langi); and Germany (River Alster in Hamburg; Graefe and Beylich, 2011) and Italy (Rota et al., 2014). There are no records from Central Europe.
Similarly, there are no exotic non-lumbricid earthworm recorded from the wild in Central Europe (except of native aquatic species Haplotaxis gorgioides and Criodrilus lacuum).
Rev. 4
On page 10 the paper states "... L. rubellus decreased significantly in degraded fen sites." This is not supported by the data in Table 4.
Authors’ Response
Thank you, we corrected that the mean density of L.rubellus increase in degraded conditions.
Rev. 4.
Generally, I did not find the results of the multivariate analyses very useful. Figures 3 and 5 were really difficult for me to interpret. If the authors want to retain those, they should be clarified.
Some of the significant p-values in Table 6 were not bolded.
Authors’ Response
We assume that visualizing of the correlation of species and their functional traits scores to centroids of the examined various hydrologic conditions and changes in groundwater level allow for better interpreting relative direction of changes in the species and trait composition change.
Rev. 4.
I wonder if some of the comparisons between the taxonomic analysis and functional analysis can be made more explicit by placing the comparable data in the same table. For example, could tables 5 and 7 be combined?
Authors’ Response
We decided to left table 5 and table 7 separately as the taxonomic and trait-based approach is presented in different sections of the results chapter.
Rev. 4
Discussion
The part about the loss of alder trees surprised me. I suggest introducing this as a possible correlate of fen drainage in the introduction.
Authors’ Response:
Our earlier study showed disappearance/loss of Caltho-Alnetum associations as a result of drainage e.q. [Nicia P., Bejger R., SterzyÅ„ska M., Zadrożny P. 2020. Passive Restoration of the Mountain Fens of the Caltho‑Alnetum Community in the Babia Góra National Park. Geomatics and Environmental Engineering 14. (2)].
Rev. 4
On page 18 the primary author of the paper is referenced using "personal communication." Shouldn't this be "personal observation?"
Authors’ Response:
Done
Rev. 4
The authors should speculate about the possible functional significance of the prostomium trend they saw. Why might it be better to have a tanylobic prostomium in degraded soils? If there is no logical reason, then this must simply be reflecting the species differences and not really functional differences.
Authors’ Response:
We add our explanation “The type of prostomium is a trait related to diet and feeding behaviur of earthworms. Tanylobic prostomium represents the most maneuverable one and enables worms to manipulate plant debris. The species with tanylobic prostomium are not restricted to one ecological group of earthworms, but they mainly include epigeic and anecic earthworms. They prevalence in degraded sites could reflects both harscher conditions in mineral soil layer (lower soil moisture) or lower content and availability of humified and fine particulate organic matter in deeper soil horizons for engogeic and amphibious worms with other prostomium types (zygolibic and epilobic) feeding on”. it.”
Reviewer 5 Report
This is an impressive work and I shall express my admiration for this scientific endeavour.
1: 45 "was the main process sorted earthworm species and traits" -> "was the main process sorting earthworm according to species and traits"
1: 49 "earthworm species sorting" -> "earthworm species filtering". You have used the term "environmental filtering" which is the ecological process filtering the potential community into a realized community.
2: 11 "soil invertebrate community diversity and structure" -> the community structure include the diversity and the abundance of populations
2: 15 "physical habitat template” -> "physical habitat". Please delete template, it is not needed in the present context. However, if you refer to the "physical habitat template" as an established scientific term, this could be a reason to maintain it.
2: 15 Please explain the type of connectivity. Just one word would be helpful.
2: 18 "deeper submergence" please explain. Will animals go deeper down in the soil and become submerged or is submergence occuring in a deeper soil horizon. Under flooded conditions submerging can occur at the soil surface, i.e. not deep.
2: 23-28 Please divide the sentence, to make it more clear.
1: 26 "regime requires a taxonomic and also functional approach" please specify more clearly why both a taxonomic and a functional approach is needed.
2: 39 "community: weighted response traits are those that are relevant to scaling the community response to environmental change" how can you exclude the effect traits when you are scaling up to the ecosystem level?
4: 10 "to hydrologic changes” -> "to hydrologic changes (HC)"
4: 11 "hydrologic change” -> "HC"
4: 12 "earthworm species” -> "earthworm communities"
4: 29 Please add or confirm that the "300 and 900 mm of water" corresponds to 30-90% water saturation or maybe you would indicate this as the water table being between 300-900 mm in a cubic meter of organic matter.
5: 23 I was puzzled about the two related concepts of "hydrologic condition" and "hydrologic alteration". Please write "hydrologic condition (HC)" instead of "hydrologic alteration" to streamline the terminology. However, as you probably deliberately uses "hydrologic alteration" please confirm that you use it in the sense of "significant changes in the magnitude, duration, timing, frequency or rate-of-change of natural stream flows" or provide another definition. On top of this you also use the expression "hydrologic conditions alteration", which may be substituted by "hydrologic alteration".
5: 26 "GWL – values of mean ground water level" is mentioned in the legend, however it is not included in the table.
5: 30-31 Please explain how you consider the enumeration, 1 to 9, as an index of the HC. Does this correspond the hydrologic alteration indices in the literature?
5: 48 You must specify the positioning of the three block including the distance between then, as well as how they were positioned at each sampling occasion.
6: 1 "soil to a depth of approximately 20–30 cm was dug" please include where these soil samples were taken and how many per study site.
6: 2 "soil samples" were they the soil block from 0 to 10 cm or were they the hand sorted soil samples to 20: 20 cm depth.
6: 24 "All five biological traits" There are 4 not 5 traits in table 2.
7: 47 You had 9 levels of HC and a unique site for each HC level, so I assume you used the three classes of HC and not the HC levels themselves (9) as a fixed factor. With 9 levels nesting would not be possible.
8: 10 "constrained by the environmental variables HC on 3 levels": Do you mean the "environmental variable HC" or the "environmental variables and HC".
8: 8-45 In this long paragraph it is not clear how much of the statistics is performed based on recommendations and guidance from the literature and which part was created by the authors.
9: 47 and Table 4psPlease use standard error of the mean instead of SD "deviation units (SD)"
Figure 1. Explain the meaning of the numbers 1 to 19 or delete them.
Table 4. Remove all the statistical details from the legend. This pertains to the data analysis section.
Please be careful when using all these trait terms of trait types: response : , functional : , ecological preference : , biological : , performance : , life: history traits and ecological traits. Biological traits of table 2 should be split into morphological and ecological traits.
11: 22 "relief": make sure to use LT instead of using another term for local topography.
11: 38 "LT group" did you define these groups? If you have created a sort of LT index lumping together the constituent LT variables, then please clarify this. It is not always clear if all levels and variables was analysed of if they were grouped like for the nine HC levels being grouped into the three disturbance levels.
11: 44 "effect on Lumbricidae community taxonomic composition variability" do you really mean variability here? Or do you refer to variability as the trends appearing after the RDA? I assume that the community composition was not entered into the analysis as a community metric but the population densities of individual earthworm species was data input to the RDA.
13: 2 "taxonomy based earthworm community composition" and the statement in 12: 24: 26 is not evident from Fig. 3. I.e. the two earthworm species is not clearly linked to the Venn diagram. Is there a resulting RDA data list of the "taxonomy based earthworm community composition"
13: 14 The table legend contains information belonging to the data analysis section. Please adjust all legends to exclude statistical analysis details.
13: 35 The "Vertical distribution" is an ecological trait.
14: 27 Please provide a reference to the way you have used the Venn diagram.
14: 32 "semi: natural fen conditions" do you actually mean "semi: natural fen HC conditions"? Please clarify.
14: 43 I believe that you also included morphological and fitness traits in the RDA analysis so "functional traits in Lumbricidae" should be "Lumbricidae traits".
22: 26-27 "Data Availability Statement" Please upload your data to a public repository. This could be Edaphobase or other databases with a quality standardized way of hosting earthworm data. It is a duty of scientists to share data used in publications and to increase transparency of the studies. Alternatively, data must be made available in the supplementary material (supporting files) section together with the trait assignments to species.
18: 3 "on the ecotype" -> "of the ecotype"; "a priori" -> "a priori"; "ecotype" change to "life: form" an ecotype refers to a genetically distinct geographic variety
18: 19 "most of individuals" -> "most individuals"
18: 21 The occurrence of lineages must be supported by COI barcodes.
20: 40 When you refer to [88] the reader cannot be sure whether your conclusion is your own or if this conclusion originated from reference [88].
The conclusion must be shortened to about half or less of the present length. If the information you must exclude from the conclusion section to shorten it is important, you must move it to the discussion. You should limit the conclusion to address the hypotheses on page 4 lines 8-13 and only include additional text if it has the nature of general conclusions and novelties obtained in the study.
Author Response
Rev. 5.
Conclusion
I suggest breaking this into two paragraphs.
Authors’ Response:
We have divided the conclusion into two paragraphs
Rev. 5
This is an impressive work and I shall express my admiration for this scientific endeavour.
Authors’ Response:
Thank you
Rev. 5
1: 45 "was the main process sorted earthworm species and traits" -> "was the main process sorting earthworm according to species and traits"
Authors’ Response:
Done
Rev. 5
1: 49 "earthworm species sorting" -> "earthworm species filtering". You have used the term "environmental filtering" which is the ecological process filtering the potential community into a realized community.
Authors’ Response:
Thank you, we corrected
Rev. 5
2: 11 "soil invertebrate community diversity and structure" -> the community structure include the diversity and the abundance of populations
Authors’ Response:
We corrected
Rev. 5
2: 15 "physical habitat template” -> "physical habitat". Please delete template, it is not needed in the present context. However, if you refer to the "physical habitat template" as an established scientific term, this could be a reason to maintain it.
Authors’ Response:
Done
2: 15 Please explain the type of connectivity. Just one word would be helpful.
Authors’ Response:
We relate to the habitat connectivity
Rev. 5
2: 18 "deeper submergence" please explain. Will animals go deeper down in the soil and become submerged or is submergence occurring in a deeper soil horizon. Under flooded conditions submerging can occur at the soil surface, i.e. not deep.
co-occurrence and also strongly limits the problems of rarity: the co-occurrence of
species’ functional features.
Authors’ Response:
We meant “submergence occurring in a deeper soil horizons”
Authors’ Response:
Rev. 5
2: 23-28 Please divide the sentence, to make it more clear.
Authors’ Response:
We change the fragment “..Therefore, to improve our understanding of how the soil biota community, adapted to a naturally disturbance-driven system, change under shifting of the disturbance regime, requires a taxonomic and also functional approach [16, 17]. Combining these two approaches, that defines species in terms of diversity patterns, their ecological roles and interactions with the environment, allow for scale up from the species to the community level”
Rev. 5
1: 26 "regime requires a taxonomic and also functional approach" please specify more clearly why both a taxonomic and a functional approach is needed.
Authors’ Response:
We explain that “Combining these two approaches, that defines species in terms of diversity patterns, their ecological roles and interactions with the environment, allow for scale up from the species to the community level”
Rev. 5
2: 39 "community: weighted response traits are those that are relevant to scaling the community response to environmental change" how can you exclude the effect traits when you are scaling up to the ecosystem level?
Authors’ Response:
Thank you for comments: e.g earthworm burrowing is as an effect trait. Therefore we change the fragment “..Within a functional trait framework, comprising response-and-effect traits [17], community-weighted response traits are those that are relevant to scaling the community response to environmental change while effect traits reflecting that change on ecosystem processes”.
Rev. 5
4: 10 "to hydrologic changes” -> "to hydrologic changes (HC)"
Authors’ Response:
Done
Rev. 5
4: 11 "hydrologic change” -> "HC"
Authors’ Response:
Done
Rev. 5
4: 12 "earthworm species” -> "earthworm communities"
Authors’ Response:
Done
Rev. 5
4: 29 Please add or confirm that the "300 and 900 mm of water" corresponds to 30-90% water saturation or maybe you would indicate this as the water table being between 300-900 mm in a cubic meter of organic matter.
Authors’ Response:
We corrected as “One cubic meter of fen organic soil retains between 300 and 900 dcm3 “.
Rev. 5
5: 23 I was puzzled about the two related concepts of "hydrologic condition" and "hydrologic alteration". Please write "hydrologic condition (HC)" instead of "hydrologic alteration" to streamline the terminology. However, as you probably deliberately uses "hydrologic alteration" please confirm that you use it in the sense of "significant changes in the magnitude, duration, timing, frequency or rate-of-change of natural stream flows" or provide another definition. On top of this you also use the expression "hydrologic conditions alteration", which may be substituted by "hydrologic alteration".
Authors’ Response:
Rev. 5
5: 26 "GWL – values of mean ground water level" is mentioned in the legend, however it is not included in the table.
Authors’ Response:
We remove GLW from the captions in the Table 1; G.L.W. mean values are presented in the Table 3.
Rev. 5
5: 30-31 Please explain how you consider the enumeration, 1 to 9, as an index of the HC. Does this correspond the hydrologic alteration indices in the literature?
Authors’ Response:
No, this is numbering of examined sites
Rev. 5
5: 48 You must specify the positioning of the three block including the distance between then, as well as how they were positioned at each sampling occasion.
Authors’ Response:
At each sampling occasion we took 3 soil monoliths (PLEASE ADD distance between blocks and how they were positioned ?)
6: 1 "soil to a depth of approximately 20–30 cm was dug" please include where these soil samples were taken and how many per study site.
6: 2 "soil samples" were they the soil block from 0 to 10 cm or were they the hand sorted soil samples to 20: 20 cm depth.
Authors’ Response:
We change the fragment – “..Three random soil blocks from the top layer (each 1/16 m2 in area and 10 cm deep) have been hand digging at each site and at each sampling occasion. In addition, below the upper soil sample the deeper soil layer immediately was dug down to a depth of approximately 20–30 cm. Earthworms living in this deeper layer were hand-sorted on site. Earthworms from the top layer were transported to the laboratory and heat-extracted using the modified Kempson extraction apparatus for a minimum of 10 days”.
Rev. 5
6: 24 "All five biological traits" There are 4 not 5 traits in table 2.
Authors’ Response:
No, we are considering 5 biological traits. Table 2 shows combinations of length max and length min traits. These traits are treated separately in the analyses.
Rev. 5
7: 47 You had 9 levels of HC and a unique site for each HC level, so I assume you used the three classes of HC and not the HC levels themselves (9) as a fixed factor. With 9 levels nesting would not be possible.
Authors’ Response:
Yes, we used 3 classes of HC as a fixed factor
Rev. 5
8: 10 "constrained by the environmental variables HC on 3 levels": Do you mean the "environmental variable HC" or the "environmental variables and HC".
Authors’ Response
We corrected – “Environmental variable HC”
Rev. 5
8: 8-45 In this long paragraph it is not clear how much of the statistics is performed based on recommendations and guidance from the literature and which part was created by the authors.
Authors’ Response
We rephrase the fragment “… We performed the univariate and multivariate analyses using the data matrix with species x fen site with the row data (individuals summed from 3 soil samples) collected at each of the fen site during the same sampling occasion and then standardized to individuals per square meter and averaged per sample. In the analysis, time [season (spring and autumn) and year of study (2010, 2011 and 2012)] was treated as a repeated measures for adjusting p-values as a function of correlation in the data”.
Rev. 5
9: 47 and Table 4psPlease use standard error of the mean instead of SD "deviation units (SD)"
Authors’ Response
We left SD - to show dispersion of the individual earthworm abundance to the mean.
Rev. 5
Figure 1. Explain the meaning of the numbers 1 to 19 or delete them.
Authors’ Response
Thank you, we corrected fig. 1 and site numbering
Rev. 5
Table 4. Remove all the statistical details from the legend. This pertains to the data analysis section.
Authors’ Response
We agree and corrected the table 4 captions
Rev. 5
Please be careful when using all these trait terms of trait types: response : , functional : , ecological preference : , biological , life: history traits and ecological traits. Biological traits of table 2 should be split into morphological and ecological traits.
Authors’ Response
We distinguished groups of traits according to Violle at. al. 2007
Rev. 5
11: 22 "relief": make sure to use LT instead of using another term for local topography.
Authors’ Response
Done
Rev. 5
11: 38 "LT group" did you define these groups? If you have created a sort of LT index lumping together the constituent LT variables, then please clarify this. It is not always clear if all levels and variables was analysed of if they were grouped like for the nine HC levels being grouped into the three disturbance levels.
Authors’ Response
We define LT group in Material and Method section; Data Analysis. Because it was not clear we rephrase the fragment.
Rev. 5
11: 44 "effect on Lumbricidae community taxonomic composition variability" do you really mean variability here? Or do you refer to variability as the trends appearing after the RDA? I assume that the community composition was not entered into the analysis as a community metric but the population densities of individual earthworm species was data input to the RDA
Authors’ Response
Yes, the population densities of individual earthworm species was data input to the RDA.
Rev. 5
13: 2 "taxonomy based earthworm community composition" and the statement in 12: 24: 26 is not evident from Fig. 3. I.e. the two earthworm species is not clearly linked to the Venn diagram. Is there a resulting RDA data list of the "taxonomy based earthworm community composition"
Authors’ Response
Thank you, we corrected the figure location in the text
Rev. 5
13: 14 The table legend contains information belonging to the data analysis section. Please adjust all legends to exclude statistical analysis details.
Authors’ Response
We corrected table captions
Rev. 5
13: 35 The "Vertical distribution" is an ecological trait.
Authors’ Response
Thank you, we move “vertical distribution “ to the ecological traits in the Table 6
Rev. 5
14: 27 Please provide a reference to the way you have used the Venn diagram.
Authors’ Response
We change the caption of the figures 2 and 4 and fragment in the data analysis section “..Variation partitioning by partial redundancy analysis (pRDA) was performed “ where the reference is cited [Borcard et al 1992].
Rev. 5
14: 32 "semi: natural fen conditions" do you actually mean "semi: natural fen HC conditions"? Please clarify.
Authors’ Response
We related to semi- natural fen HC conditions
Rev. 5
14: 43 I believe that you also included morphological and fitness traits in the RDA analysis so "functional traits in Lumbricidae" should be "Lumbricidae traits".
Authors’ Response
Yes, we change
Rev. 5
22: 26-27 "Data Availability Statement" Please upload your data to a public repository. This could be Edaphobase or other databases with a quality standardized way of hosting earthworm data. It is a duty of scientists to share data used in publications and to increase transparency of the studies.
Alternatively, data must be made available in the supplementary material (supporting files) section together with the trait assignments to species.
Authors’ Response
Yes, with revised version of the MS we upload the supplementary material
Rev. 5
18: 3 "on the ecotype" -> "of the ecotype"; "a priori" -> "a priori"; "ecotype" change to "life: form" an ecotype refers to a genetically distinct geographic variety
Authors’ Response
Done
Rev. 5
18: 19 "most of individuals" -> "most individuals"
Authors’ Response
Done
Rev. 5
18: 21 The occurrence of lineages must be supported by COI barcodes.
Authors’ Response
We change – that group
Rev. 5
20: 40 When you refer to [88] the reader cannot be sure whether your conclusion is your own or if this conclusion originated from reference [88].
The conclusion must be shortened to about half or less of the present length. If the information you must exclude from the conclusion section to shorten it is important, you must move it to the discussion. You should limit the conclusion to address the hypotheses on page 4 lines 8-13 and only include additional text if it has the nature of general conclusions and novelties obtained in the study.
Authors’ Response
We agree and we change the fragment “…This result leads to the conclusion that the spatial and temporal heterogeneity in the physicochemical environment of mountain fens is one of the dimensions on which characteristic species traits can be sorted. Our finding support the Townsed and Hildrew hypothesis of species traits relation to habitat templet [88]”.
Round 2
Reviewer 1 Report
Authors have significantly improved the manuscript addressing all the issues raised by me and the other reviewers, and I believe that now the manuscript is ready for publication virtually as it is.
I have only noticed a few editorial issues, such as lack of spaces between the text and Tables and/or Table Captions, For instance see Table 2 which lacks spaces both between the text preceding the Table Caption and following the Table, but other Tables and Figures should also be checked. But this is probably an issue which should be resolved by the editor during the final text editing.
Reviewer 5 Report
My coments were all addressed
dcm3 -> dm3